# Distress and Wellbeing among General Practitioners in 33 Countries during COVID-19: Results from the Cross-Sectional PRICOV-19 Study to Inform Health System Interventions

**DOI:** 10.3390/ijerph19095675

**Published:** 2022-05-06

**Authors:** Claire Collins, Els Clays, Esther Van Poel, Joanna Cholewa, Katica Tripkovic, Katarzyna Nessler, Ségolène de Rouffignac, Milena Šantrić Milićević, Zoran Bukumiric, Limor Adler, Cécile Ponsar, Liubove Murauskiene, Zlata Ožvačić Adžić, Adam Windak, Radost Asenova, Sara Willems

**Affiliations:** 1Research Centre, Irish College of General Practitioners, D02 XR68 Dublin, Ireland; 2Department of Public Health and Primary Care, Ghent University, 9000 Ghent, Belgium; els.clays@ugent.be (E.C.); esther.vanpoel@ugent.be (E.V.P.); sara.willems@ugent.be (S.W.); 3Institute of Health and Society, Catholic University of Louvain, 1348 Louvain-la-Neuve, Belgium; joanna.cholewa@student.uclouvain.be (J.C.); segolene.derouffignac@uclouvain.be (S.d.R.); cecile.ponsar@uclouvain.be (C.P.); 4City Institute of Public Health Belgrade, 11000 Belgrade, Serbia; katica.tripkovic@zdravlje.org.rs; 5Department of Family Medicine, Jagiellonian University Medical College, 31-061 Krakow, Poland; katarzynanessler@gmail.com (K.N.); mmwindak@cyf-kr.edu.pl (A.W.); 6Faculty of Medicine, University of Belgrade, 11000 Belgrade, Serbia; milena.santric-milicevic@med.bg.ac.rs (M.Š.M.); zoran.bukumiric@med.bg.ac.rs (Z.B.); 7Department of Family Medicine, Sackler Faculty of Medicine, Tel Aviv University, Tel Aviv 6997801, Israel; limchuk@gmail.com; 8Public Health Department, Faculty of Medicine, Vilnius University, LT-01513 Vilnius, Lithuania; murauskiene@mtvc.lt; 9Department of Family Medicine, School of Medicine, University of Zagreb, 10000 Zagreb, Croatia; zlata.ozvacic@mef.hr; 10Department of General Practice, Medical University Plovdiv, 4003 Plovdiv, Bulgaria; radost_assenova@yahoo.com

**Keywords:** wellbeing, distress, COVID-19, general practice/family medicine, health system, organizational, interventions

## Abstract

Emerging literature is highlighting the huge toll of the COVID-19 pandemic on frontline health workers. However, prior to the crisis, the wellbeing of this group was already of concern. The aim of this paper is to describe the frequency of distress and wellbeing, measured by the expanded 9-item Mayo Clinic Wellbeing Index (eWBI), among general practitioners/family physicians during the COVID-19 pandemic and to identify levers to mitigate the risk of distress. Data were collected by means of an online self-reported questionnaire among GP practices. Statistical analysis was performed using SPSS software using Version 7 of the database, which consisted of the cleaned data of 33 countries available as of 3 November 2021. Data from 3711 respondents were included. eWBI scores ranged from −2 to 9, with a median of 3. Using a cutoff of ≥2, 64.5% of respondents were considered at risk of distress. GPs with less experience, in smaller practices, and with more vulnerable patient populations were at a higher risk of distress. Significant differences in wellbeing scores were noted between countries. Collaboration from other practices and perception of having adequate governmental support were significant protective factors for distress. It is necessary to address practice- and system-level organizational factors in order to enhance wellbeing and support primary care physicians.

## 1. Introduction

The COVID-19 pandemic has undeniably impacted physical and mental health across all population groups. However, healthcare workers seem to be one of the most vulnerable groups due to the high risk of infection, increased work stress, and fear of spreading the infection to their families [1,2,3]. Emerging literature from all around the world and media reports highlight the huge toll frontline health workers pay to the COVID-19 pandemic. However, prior to the crisis, this group was already identified as fragile, and their mental health and wellbeing were already major health issues [1,4]. Burnout, which reached epidemic levels among healthcare providers well before this health crisis, is the most extreme form of this lack of wellbeing [5]. Symptoms presented by healthcare workers include depression, depersonalization, emotional exhaustion, a sense of reduced personal accomplishment, anxiety, stress, and cognitive and social problems. These symptoms, when they occur, not only have a direct impact on the physician but also on patients. They affect physicians’ own care and safety and professionalism, with, as a corollary, a deleterious effect on the quality of care, the safety of patients, and healthcare access. As such, they jeopardize the sustainability of the healthcare system [3,5,6].

Some studies have shown that among physicians who report experiencing at least some signs of burnout, family medicine and emergency medicine physicians are among those at highest risk [7]. A high-stress role combined with an even more anxiety-provoking and deleterious work environment caused by the COVID-19 pandemic have undoubtedly exposed general practitioners/family practitioners (GPs), as first-line workers, to a higher risk of developing distress. While some researchers have shown that one way to reduce the risk of burnout and promote wellbeing in the medical profession is to address not only the individual but also the work environment [3,5], studies focus more on reducing the negative aspects of stress rather than strengthening the positive aspects by finding ways to promote wellbeing [2,6,8]. Moreover, little is known about the impact of the pandemic on the wellbeing of the population of GPs. GPs are well known to manage their own physical and mental health and often try to respond positively to the implicit expectation of their patients and colleagues to always look their best, even when sick, as their health is often interpreted as an indicator of their medical competence [9].

The aim of this paper is to measure the frequency of distress and wellbeing among general/family practitioners during the COVID-19 pandemic and to identify some of the key levers that could potentially mitigate the risk of distress.

## 2. Materials and Methods

### 2.1. Study Design and Setting

In the summer of 2020, an international consortium of more than 45 research institutes was formed under the coordination of Ghent University (Belgium) to set up the study to consider how PRImary care practices were organized during the COVID-19 pandemic (PRICOV-19). This multi-country cross-sectional study specifically focused on quality and safety in primary care practices during the COVID-19 pandemic. Data were collected in 37 European countries and Israel. Data were collected by means of an online self-reported questionnaire among general/family practices. The questionnaire was developed at Ghent University in multiple phases, including a pilot study among 159 general practices in Flanders (Belgium). More details are described in the protocol [10]. The questionnaire consists of 53 items divided into six topics: (a) infection prevention, (b) patient flow for COVID-19 and non-COVID-19 care, (c) dealing with new knowledge and protocols, (d) communication with patients, (e) collaboration, (f) wellbeing of the respondent, and (g) characteristics of the respondent and the practice. Permission was granted to use the Mayo Clinic Wellbeing Index [10]; the expanded 9-item Wellbeing Index (eWBI) version was utilized [11]. The questionnaire was translated into 38 languages following a standard procedure. The Research Electronic Data Capture (REDCap) platform was used to host the questionnaire in all languages, send out invitations to the national samples of general/family practices, and securely store the answers from the participants [12]. Throughout this paper, ‘general practice’ is used to refer collectively to general practice and family medicine, and ‘general practitioner (GP)’ is used to refer to general practitioners and family (medicine) practitioners.

### 2.2. Sampling and Recruitment

The data reported here were collected between November 2020 and December 2021, except for Belgium, where data were partially collected earlier. Data collection varied between countries from 3 to 35 weeks. In each partner country, the consortium partner(s) recruited general/family practices following a predefined recruitment procedure [10], which is shown in the Appendix A. There was no funding for this study, and coordinators recruited practices out of goodwill; however, a randomized sample was requested where possible. One questionnaire was completed per practice, preferably by a general practitioner/family practitioner or by a staff member familiar with the practice organization. In most countries, the number of practices was unknown, although the number of individual practitioners may have been available. Given the voluntary nature of participation and that the number of practices was unknown, it was not possible to enforce either a specific recruitment strategy or specific response numbers/rate. The majority of countries choose a sample from the entire GP population, with this being a convenience sample for approximately one-half (Appendix A). At least one reminder was sent in all countries. The overall response rate was 27.8%. The response rate varied, and, generally, targeted convenience samples attracted larger response rates. However, this is not consistent with recruitment/sampling strategies, as one might expect some notable high response rates from random/convenience national samples (e.g., Bulgaria, Greece, Serbia, and Spain) (Appendix A).

### 2.3. Data Analysis

Statistical analysis was performed using SPSS software (version 28.0 SPSS Inc., Chicago, IL, USA) using Version 7 of the database, which was the version consisting of the cleaned data of 33 countries available as of 3 November 2021. Ghent University was responsible for the data cleaning of the international data. Cases missing the outcome variable of interest, the eWBI data, were excluded. Within the eWBI, seven items are responded to with a yes (scored as 1) or no (scored as 0), and the remaining two items are responded to on a 7-point or 5-point Likert scale, with those responding strongly disagree/disagree having one point added to their score, those who responded agree/strongly agree having one point subtracted from their score and no adjustment made for those with middle neutral responses. Being at risk of distress is defined as a score of ≥2, as per previous studies [11].

Linear mixed model analysis was undertaken (due to the clustering of respondent practices in countries), with the continuous eWBI score as the outcome variable. The conditions for linear mixed model analysis were met, and we checked for normality of residuals and for constant error variance (by plotting residuals against fitted values). We tested different random intercept models using restricted maximum likelihood estimation. Four models were tested using a stepwise approach with the null model (Model I) permitting the calculation of the intraclass correlation coefficient (ICC), assessing the proportion of the variance in the outcome variable that can be explained by country. In subsequent models, we added individual GP experience (Model II), practice characteristics (Model III), and COVID-19 context characteristics (Model IV) as fixed effects. A larger number of relevant variables were tested in the preliminary analysis (model exploration phase), taking into account potential multicollinearity. The Akaike’s Information Criterion (AIC) and −2 log-likelihood values were used as goodness-of-fit model criteria. The likelihood ratio test was used to compare model fit between nested models. Additionally, we ran the same models using logistic regression on the eWBI dichotomous variable of <2 and ≥2, as 2 is the standard cutoff to indicate distress [11]. We have included this data in the Appendix A.

Various country groupings were considered; however, individual country was included as a determinant. Country grouping based on geography alone [13] was not considered valid in this instance, with countries within each grouping not consistent in terms of factors such as health system structure/organization and payment system. The geographic grouping also resulted in only two countries in one group, and both health system structure factors and number of responses did not warrant this as a valid grouping. Groupings based OECD healthcare system [14] and reported strength of the primary care system [15] were also considered but were insufficient, as data on all countries included in our study was not available.

Individual GP experience was coded as a categorical variable in 10-year age groups (0–9, 10–19, 20–29, 30–39 years). The practice characteristics were the size of the practice and the practice location. Multiple variables were available for practice size—number of patients, GPs, and paid staff (with all recoded into quartiles because of highly skewed distributions). These variables were highly correlated, and based on exploratory analysis, the number of GP staff was retained in the model. We also considered the ratio of number of patients to number of GPs. The questionnaire did ask for full-time equivalent (FTE) GPs/all staff; however, due to the unreliability and inconsistency in the recording of this data, this ratio was not possible to calculate. The ratio of patients to total number of GPs in the practice (i.e., regardless of whether full or part-time) is not a good measure of workload when the FTE data are unknown due to a lack of clear interpretation of its meaning. We ran the models with this variable included and refer to the results below, although due to the above, it was not included in the final formal analysis.

Patient population composition variables: respondents were asked to what extent they felt their patient population was below, approximately at, or above the average of practices in their country in terms of treating patients with chronic conditions, patients over the age of 70, patients with limited or low health literacy, patients with a migration background with difficulty speaking the local language, patients with financial problems, patients with a psychiatric vulnerability, and patients with little social support or limited informal care. Because of high inter-relatedness, chronic disease and financial problems were retained in the model following the exploratory phase. The COVID-19-related contextual factors (i.e., the respondents’ statements regarding the effect of the COVID-19 pandemic on general practitioners’ practice in the local context) were scored on a 5-point Likert scale from 0 (strongly disagree) to 4 (strongly agree) and included collaboration (whether the practice could count on the help of other local PC practices if staff members were absent because of COVID-19), adequate support from the government for the proper functioning of practice, whether respondents’ responsibilities in this practice increased due to the pandemic the COVID-19 pandemic, whether respondents needed further training for these amended responsibilities and whether there was enough protected time provided to the GPs for reviewing guidelines or going through relevant scientific literature.

### 2.4. Ethical Approval

The study was conducted according to the guidelines of the Declaration of Helsinki. The Research Ethics Committee of Ghent University Hospital approved the protocol of the PRICOV-19 study (BC-07617). Research ethics committees in the different partner countries gave additional approval if needed in that country. All participants gave informed consent on the first page of the online questionnaire.

## 3. Results

The analysis included 3711 GPs who had a valid eWBI score. Responses from 33 countries were received (see Appendix A Appendix A). A description of the main characteristics of the sample is shown in Table 1. Approximately one-quarter of the respondents were in each 10-year age group. In terms of location, the majority (42.9%) of practices were based in cities/suburbs and were single-handed (39%) practices. With regard to the patient population, 39.1% considered that the proportion of patients with chronic conditions in their practice was above average for their country. The comparable figure for patients with financial problems was 22.3%.

Table 2 shows the results in terms of GPs’ opinions of the effects of the COVID-19 pandemic on their practice. The scale for each item was a 5-point Likert scale. Overall, 43.6% of respondents reported that they could not count on the support of other practices in their area if staff were out due to COVID-19, and 53.6% indicated that adequate support from the government for the proper functioning of the practice did not exist. Only one-third (32.9%) agreed there was enough time to review guidelines or read the relevant scientific literature.

The total eWBI scores among respondents ranged from −2 to 9, with a mean of 2.7 (SD 2.7) and a median of 3, with lower scores indicating better wellbeing and higher scores indicating higher distress (Table 3). Total scores per country varied, as shown in Figure 1. Overall, 64.5% (2394 out of 3711) of respondents had a score of ≥2 and, therefore, were considered at risk of distress.

Table 4 presents the results of the linear mixed model analysis of GPs’ distress and a set of potential individual and practice-related predictors. Model I, showing the null model, or intercept-only model, has an ICC = 16.9%, meaning that 17% of the variance in eWBI is attributable to the country. Each subsequent stepwise model shows a better goodness-of-fit (based on smaller AIC and –2 log-likelihood values). The likelihood ratio test shows that each model fits significantly better than the previous one. Variances at the group and individual levels reduce when adding predictors, except for intercept variance in Model II. There is a reduction in the group-level variance, indicating that the GP-level characteristics have compositional effects or that a large variance in the distress scores noted between countries is reduced by adding individual-level predictors. However, country individual-level variance remains significant. In an alternative analysis, the ratio of patients to GP working in the practice was a significant determinant of wellbeing in Model III but was no longer significant in Model IV when contextual factors were added.

In Model IV, GP, practice and contextual factors are independently and significantly related to eWBI score. Compared with GPs with ≥30 years of experience, all other groups had significantly higher distress scores. Compared with GPs working in large practices of at least five GPs, those working single-handed or in duo practices had higher distress. The location of the practice was not significant overall; however, a significant difference between mixed urban–rural practices compared to those in cities and suburbs was observed. GPs working with more vulnerable patient populations were at higher risk of distress. In particular, those having an average or more than average proportion of patients with financial difficulties was associated with higher distress. The experience of collaboration from practices in the neighborhood and adequate governmental support were significant protective factors for distress. Respondents perceiving an increase in responsibilities since the COVID-19 pandemic and sensing a related need for further training had higher distress scores. Having enough protected time in the GP’s schedule to review guidelines or read the scientific literature was also a protective factor. However, in this final model, both country- and individual-level variance remained significant.

Logistic regression using the same models on the binary eWBI variable using a cutoff of 2 was also performed. The results show similar results to the linear regression with the same significant factors determining ‘in distress’ as the overall eWBI continuous score. These results are included in the Appendix A. We performed the same analysis using different cutoff values (cutoff of 4 and 5), and all resulted in the same factors being significant for determining distress as when using the standard cutoff of 2 [11].

## 4. Discussion

Previous work has reported more anxiety and depression in younger groups and higher burnout on items of the burnout scale relating to patient interactions among older GPs [16]. We did not collect GP age; however, our results show that those with fewer years of experience as a GP showed higher distress.

A systematic review of 59 studies exploring the impact of COVID-19 on the mental health of professionals fighting against the pandemic [17] concluded that healthcare workers valued contact with and the social support of collaborators, resulting in less frequent mental health problems. Results from other studies showed that GPs suffered from depression and anxiety more frequently than hospital physicians, who regularly work in bigger teams [18], and that GPs working in larger practices are more satisfied and have fewer burnout symptoms than those working in single-handed practices [19]. These findings are in line with the results of our study, which showed that GPs working in larger practices had significantly lower distress than those working in solo or duo practices.

Our study showed that one’s perception of having adequate governmental support, experiencing collaboration from practices in the neighborhood, and having enough protected time to review guidelines and the scientific literature were significant protective factors for distress. Similar findings have been reported in previous studies. Perception of being protected by the state during the COVID-19 outbreak was associated with a lower presence of symptoms of generalized anxiety disorder among Columbian GPs [20]. A rapid review concluded that support and recognition from the government were also identified as factors protecting against adverse mental health [21]. Prior to COVID-19, strong management support, teamwork, and social network were previously identified as promoters of professional resilience in GPs working in challenging environments [22]. Systemic support contributes to GPs’ feeling of high morale and that their work is respected and valued, which promotes positive psychological functioning and wellbeing [23].

Professional collaboration and solidarity between GPs are considered very important in terms of emotional support during crises, such as a pandemic. This is in line with studies conducted among GPs in multiple countries [24,25,26,27] and confirmed by the results of our study. A study in Italy reported less professional experience was associated with higher levels of anxiety and depression, which corresponds with our findings [28].

In a review by Long et al., GPs with high job satisfaction reported support through good relationships at the practice, which enabled them to successfully adapt to external pressures and remain in the profession [29]. Similarly, the resilience of GPs working in areas with socioeconomic deprivation was related more to supportive teams than to the GP’s individual characteristics [30]. Encouraging collaborative relationships between general practices is needed not only to preserve GPs’ mental health but to expand the response capacity for current and future crises.

Time pressure is a common stressor in general practice and negatively affects the GP seeking relevant information and reading scientific literature [31,32]. During a pandemic, increased time pressure due to a higher workload, uncertainties regarding an unknown disease, work reorganization, and potential lack of confidence in caring for COVID-19 patients might underpin additional psychological distress among GPs [33,34,35,36,37,38,39]. Time pressure has been recognized as one of the workplace challenges with a negative impact on GPs’ professional resilience [22]. Therefore, it is important to ensure that GPs have enough time to study guidelines carefully and keep up to date with the best evidence in order to relieve their stress and improve patient safety.

Data collection took place over a relatively long period. It is a limitation that the questionnaire did not collect data on the wave and stage of COVID at the time of completion. This could have varied between countries but also within countries, particularly when data collection was over a longer period. However, it is not possible to accurately retrospectively establish the exact COVID burden at each time point in each country, and this restricts our ability to comment on the impact this may have had on wellbeing. Our surveys were based on a self-selecting sample, which comes with inherent bias. This is often referred to as volunteer bias and can be mitigated by larger sample sizes, as we had here overall. Self-selection/volunteer bias may have resulted in either higher or lower wellbeing scores, as it could be linked to motivation. However, as the main focus of the study was on the organization of primary care and the reference to wellbeing was not immediately relevant at the outset, the potential impact on wellbeing scores may be less than for other aspects covered by the study/questionnaire. Given the potential volunteer bias and the cross-sectional survey design [10], direct assessment of causal relationships was not possible. We also did not collect information on the actual support measures implemented or the requirements placed on practices; thus, what we reported on was the respondent’s perception of support/change. Exact data on the population of general practices in every partner country was not available to calculate the target sample size, and, additionally, given the volunteer nature of the study, no minimum sample size requirements were applied to participation. For this reason, we did not focus on presenting individual country-level data in detail. Given that full randomization was not possible in all countries, a sampling bias may exist, which might have affected external validity. Some strategies were implemented to minimize the potential biases encompassed by conducting multicenter surveys. Each partner undertook the translation (and back-translation) and cultural adaptation of the questionnaire first, and then after resolving terminology issues, the collaborators reached the harmonized version of the questionnaire, with consideration of local arrangements and definitions. This rigorous development of the questionnaire is a strength of the study. Another strength of the study is the large sample size, the broad scope of respondents, and the inclusion of almost all different circumstances that European primary care operates under, making the findings more generalizable.

## 5. Conclusions

GPs with less experience, in smaller practices, and with more vulnerable patient populations are at a higher risk of distress. Collaboration with other practices and perception of having adequate governmental support are significant protective factors for distress. Significant differences in wellbeing scores were noted between countries. Practice- and system-level organizational factors are needed to enhance wellbeing and support primary care physicians.

## Figures and Tables

**Figure 1 ijerph-19-05675-f001:**
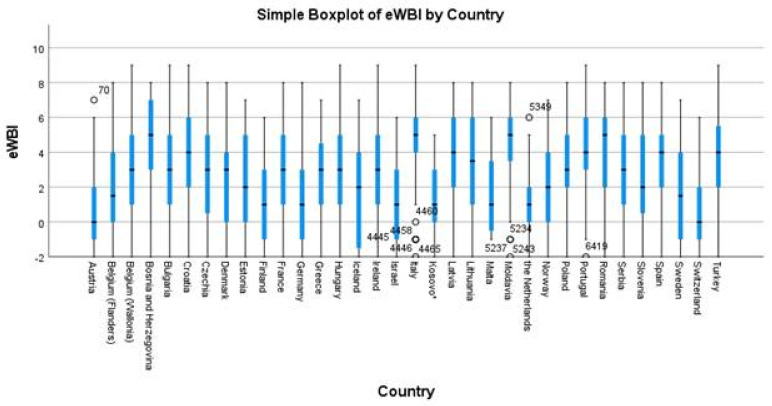
Box plot of GPs’ total eWBI scores (on a scale from −2 to 9) during the COVID-19 pandemic in 2021 by country (*n* = 3711). * All references to Kosovo, whether the territory, institutions, or population, in this project shall be understood in full compliance with the United Nations Security Council Resolution 1244 and the ICJ Opinion on the Kosovo declaration of independence, without prejudice to the status of Kosovo.

**Table 1 ijerph-19-05675-t001:** Main characteristics of the general practitioners and their practices during the COVID-19 pandemic (*n* = 3711).

		*n*	%
**GP Individual Factors**			
	**Years of experience (*n* = 3698)**		
	0–9	941	25.4
	10–19	930	25.1
	20–29	1022	27.6
	30–39	805	21.8
**Practice Factors**	**Location of practice (*n* = 3700)**		
	Big (inner) city	1215	32.8
	Suburbs	374	10.1
	(Small) town	673	18.2
	Mixed urban–rural	751	20.3
	Rural	687	18.6
	**Number of GPs (*n* = 3675)**		
	1	1433	39.0
	2	585	15.9
	3–4	742	20.2
	5+	915	24.9
	**Patients with chronic conditions (*n* = 3624)**		
	Below average	178	4.9
	Approx. average	2030	56.0
	Above average	1416	39.1
	**Patients with financial problems (*n* = 3571)**		
	Below average	814	22.8
	Approx. average	1962	54.9
	Above average	795	22.3

**Table 2 ijerph-19-05675-t002:** Respondents’ opinions of the effects of the COVID-19 pandemic on their practice.

	*n*	Strongly Disagree %	Disagree %	Neutral %	Agree %	Strongly Agree %	Mean (SD)
If staff members in this practice are absent because of COVID-19, this practice can count on the help of other PC practices in the neighborhood	3540	20.7	22.9	13.2	30.7	12.4	1.91 (1.36)
There is adequate support from government for proper functioning of practice	3643	20.0	33.6	23.4	18.7	4.3	1.53 (1.13)
Since COVID-19, my responsibilities in this practice increased	3296	2.6	4.4	15.4	32.2	45.4	3.13 (1.00)
I need further training for these amended responsibilities since COVID-19	3238	13.1	25.3	28.3	27.4	5.9	1.88 (1.13)
Since COVID-19, in this practice, there is enough protected time provided for reviewing guidelines scientific literature	3644	28.2	24.0	14.8	22.3	10.6	1.63 (1.37)

**Table 3 ijerph-19-05675-t003:** GPs’ eWBI components and total scores during the COVID-19 pandemic (*n* = 3711).

		*n*	%
During the past month, have you felt burned out from your work?	No	1242	33.5
	Yes	2469	66.5
During the past month, have you worried that your work is hardening you?	No	1638	44.1
	Yes	2073	55.9
During the past month, have you often been bothered by feeling down, depressed, or hopeless?	No	2083	56.1
	Yes	1628	43.9
During the past month, have you fallen asleep while sitting inactive in a public place?	No	3322	89.5
	Yes	389	10.5
During the past month, have you felt that all the things you had to do were piling up so high that you could not overcome them?	No	1754	47.3
	Yes	1957	52.7
During the past month, have you been bothered by emotional problems (such as feeling anxious, depressed, or irritable)?	No	1589	42.8
	Yes	2122	57.2
During the past month, has your physical health interfered with your ability to do your daily work at home and/or away from home?	No	2481	66.9
	Yes	1230	33.1
The work I do is meaningful to me.	1 (Strongly disagree)	35	0.9
	2	54	1.5
	3	123	3.3
	4	222	6.0
	5	684	18.4
	6	987	26.6
	7 (Strongly agree)	1606	43.3
My work schedule leaves me enough time for my personal/family life.	1 (Strongly disagree)	755	20.3
	2	938	25.3
	3	907	24.4
	4	750	20.2
	5 (Strongly agree)	361	9.7
eWBI scores	−2	310	8.4
	−1	291	7.8
	0	331	8.9
	1	385	10.4
	2	414	11.2
	3	427	11,5
	4	428	11.5
	5	444	12.0
	6	419	11.3
	7	191	5.1
	8	59	1.6
	9	12	0.3
Mean: 2.7; SD: 2.7; Median: 3			

**Table 4 ijerph-19-05675-t004:** Results of linear mixed model analysis of potential predictors for GPs’ distress(total eWBI score) during the COVID-19 pandemic.

Linear Mixed Models, Fixed Effect Estimate (95% CI) for Total eWBI Score				
	Model I: Fixed Effect Estimate (95% CI)	Model II: Fixed Effect Estimate (95% CI)	Model III: Fixed Effect Estimate (95% CI)	Model IV: Fixed Effect Estimate (95% CI)
Intercept	2.65 (2.24, 3.07) ***	2.04 (1.59, 2.50) ***	0.91 (0.30, 1.53) **	1.00 (0.32, 1.68) **
GP experience				
30–39 years		Ref.	Ref.	Ref.
20–29 years		0.69 (0.45, 0.93) ***	0.74 (0.50, 0.99) ***	0.50 (0.25, 0.76) ***
10–19 years		0.73 (0.49, 0.98) ***	0.85 (0.60, 1.10) ***	0.54 (0.28, 0.80) ***
0–9 years		0.81 (0.56, 1.06) ***	0.87 (0.62, 1.12) ***	0.48 (0.21, 0.74) ***
Number of GP staff in practice				
≥5			Ref.	Ref.
3–4			0.22 (−0.05, 0.49)	0.19 (−0.08, 0.47)
2			0.35 (0.06, 0.65) *	0.35 (0.05, 0.65) *
1			0.50 (0.22, 0.78) ***	0.56 (0.27, 0.86) ***
Practice location				
Big (inner) city			Ref.	Ref.
Suburbs			0.09 (−0.22, 0.40)	0.03 (−0.29, 0.34)
(Small) town			0.17 (−0.09, 0.42)	0.07 (−0.19, 0.33)
Mixed urban–rural			0.40 (0.16, 0.65) **	0.29 (0.04, 0.55) *
Rural			0.07 (−0.18, 0.33)	0.13 (−0.14, 0.39)
Patient population: chronic disease				
Below average			Ref.	Ref.
Approximately average			0.07 (−0.33, 0.47)	0.01 (−0.40, 0.41)
Above average			0.66 (0.25, 1.07) **	0.47 (0.05, 0.89) *
Patient population: financial problems				
Below average			Ref.	Ref.
Approximately average			0.40 (0.19, 0.62) ***	0.34 (0.11, 0.56) **
Above average			0.69 (0.43, 0.95) ***	0.59 (0.33, 0.86) ***
Collaboration from neighborhood practices (0–4)				−0.17 (−0.24, −0.10) ***
Adequate government support (0–4)				−0.31 (−0.40, −0.23) ***
Responsibilities have increased (0–4)				0.39 (0.30, 0.48) ***
Further training for amended responsibilities needed (0–4)				0.36 (0.27, 0.44) ***
Enough protected time for reviewing guidelines/literature (0–4)				−0.33 (−0.40, −0.26) ***
Intercept variance (s.e.)	1.31 (0.35) ***	1.35 (0.36) ***	1.26 (0.33) ***	0.49 (0.15) ***
Residual variance (s.e.)	6.42 (0.15) ***	6.32 (0.15) ***	6.11 (0.15) ***	5.31 (0.14) ***
MODEL INFORMATION				
Akaike’s Information Criterion (AIC)	17,535.32	17,423.17	16,262.10	12,889.80
−2 log likelihood	17,531.32	17,419.17	16,258.10	12,885.80
Likelihood ratio test		112.15 (df = 3) ***	1161.07 (df = 11) ***	3372.30 (df = 16) ***

* *p* < 0.05; ** *p* < 0.01; *** *p* < 0.001.

## Data Availability

The anonymized data is held at Ghent University and is available to participating partners for further analysis upon signing an appropriate usage agreement.

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
