# Peer review of "Distress and Wellbeing among General Practitioners in 33 Countries during COVID-19: Results from the Cross-Sectional PRICOV-19 Study to Inform Health System Interventions"

_ijerph, 2022, doi:10.3390/ijerph19095675_

Round 1

Reviewer 1 Report

This is a well-written manuscript on COVID-19 related distress among general practitioners. The introduction provides sufficient data for the study, which was conducted as an online self-reporting questionnaire among GPs in 33 countries. The methodology is appropriate and described in detail. The data are logically and systematically presented. The discussion is informative and well-balanced, discussing the study's data in the context of previous findings. The weaknesses and strengths of the study are recognized and openly presented. This is a worthy study defining the scope of the issue and identifying protective mechanisms to preserve the wellbeing of GPs at the time of the pandemic. I suggest only minor text editing before publishing. 

Author Response

Thank you for your comments. We have undertaken a spell check as suggested.

Reviewer 2 Report

Thank you for the opportunity to review “Distress and wellbeing among general practitioners in 33 countries during COVID-19: results from the cross-sectional PRICOV-19 study to inform health system interventions”. This manuscript   examined the distress and wellbeing, as well as associated factors, among 3,711 general practitioners from 33 countries and regions during the pandemic of COVID-19. This study is interesting and also relevant. The authors concluded that GPs with less experience, in smaller practices, and with more vulnerable patient populations are at a higher risk of distress, which is reasonable as expected. The authors also recommend that adequate governmental support, experiencing collaboration from practices in the neighborhood, and having enough protected time to review guide-lines and the scientific literature are significant protective factors for distress. The manuscript was mostly clear with specific issues noted below.

Major:

  1. As the study was conducted over a time period between November 2020 and December 2021, covering the peak period and winding down period of the pandemic, the perception of its risk and physician’s distress are very like to differ by the time when they were surveyed. I would suggest that the author should account for different survey time.
  2. It was not clear how the sampling was done. The author stated in sampling and recruitment section “PRICOV-19 aimed to sample between 80 and 200 practices per country, depending on the total number of practices.”  However, it is not very clear how the subjects in each country were samples, ie. Were they clustered in one city or a few places or the same practice group? Given this small sample size (80-200) and unclear sampling scheme, the generalizability of the findings is questionable.
  3. This study lacks sensitivity/counterfactual analysis. I suggest that the authors redo Model IV for different subgroups of the surveyed individual practitioners to validate their discussion/conclusion.
  4. It is well studies that distress level is also associated with the patient volume. The author argued that it was highly corrected with number of staff. If this was the case, had the authors consider including the average number of patients per physician and number of staff?
  5. I also notice that a cut-off at 2 for the eWBI score is discussed as a criterion of risk of distress. I am curious why a mixed effect model with a binary response was not considered to examine the relationship between risk factors and distress. The estimate (i.e. odd ratio, or relative risk) would be more interpretable.
  6. In the discussion, the authors mention the self-selection bias in this study. Other than increasing the sample size, I think the authors could consider other bias-reduction techniques in this study.
  7. 7. Support from government and requirement/ of amended responsibilities during the training, and guideline for practice during COVID could vary by country. These are also potential confounder.  However, it was neither discussed nor acknowledged as a limitation.

Minor concerns:

  1. Please spell out the full name of PRICOV-19 when the first time it appeared.
  2. How many subjects with missing value of key covariates was not described at all.
  3. When talking about ICC, the authors mentioned “country clustering”, which confuses me. In my opinion, by “country clustering” we mean to sample several countries among all available countries
  4. The response variable eWBI score is integer-valued with a range -2 to 9. Further analysis may be needed to clarify why a linear mixed effect model, which is supposed to be applied for continuous variable, is appropriate for this integer-valued response variable.
  5. The boxplots in Figure 1 shows skewness for some countries, for example, Austria and Denmark. This is again an evidence that the appropriateness of the linear mixed effect model needs to be validated.

Author Response

Major:

  1. As the study was conducted over a time period between November 2020 and December 2021, covering the peak period and winding down period of the pandemic, the perception of its risk and physician’s distress are very like to differ by the time when they were surveyed. I would suggest that the author should account for different survey time.

 This is an aspect which has been discussed extensively by the author group. The COVID wave/stage was not collected as part of the study, which is a limitation, It is not possible to accurately retrospectively establish the exact COVID burden at each time point in each country. We recognised this is a limitation and we have now referred to in the limitations section.

  1. It was not clear how the sampling was done. The author stated in sampling and recruitment section “PRICOV-19 aimed to sample between 80 and 200 practices per country, depending on the total number of practices.”  However, it is not very clear how the subjects in each country were samples, ie. Were they clustered in one city or a few places or the same practice group? Given this small sample size (80-200) and unclear sampling scheme, the generalizability of the findings is questionable.

We have added additional detail to the Sampling/Recruitment section and also expanded the supplementary table to include the recruitment/sampling strategy used in each country and the related response rate. We have additionally commented on this in the limitations section.

  1. This study lacks sensitivity/counterfactual analysis. I suggest that the authors redo Model IV for different subgroups of the surveyed individual practitioners to validate their discussion/conclusion.

The only individual level variable is GP years of experience and this is included as a key determinant, as supported by the literature. This is no other individual factor on which we can conduct such a sub-group analysis. Hence we have not made any changes with regard to this comment. However, if we have mis-understood your suggestion, please do advise and we will be happy to consider how we can address it.

  1. It is well studies that distress level is also associated with the patient volume. The author argued that it was highly corrected with number of staff. If this was the case, had the authors consider including the average number of patients per physician and number of staff?

Thank you for this suggestion. We have looked at the ratio of number of patients to number of GPs. The questionnaire did ask for FTE GPs/all staff, however, due to the unreliability and inconsistency in the recording of this data, this ratio was not possible to calculate. The ratio of patients to total number of GPs in the practice (i.e. regardless of whether full or part-time) is not a good measure of workload per doctor. However, we ran the models with this variable included and have included a commentary about this in the methods section, although due to the above, it is not included in the final formal analysis.

  1. I also notice that a cut-off at 2 for the eWBI score is discussed as a criterion of risk of distress. I am curious why a mixed effect model with a binary response was not considered to examine the relationship between risk factors and distress. The estimate (i.e. odd ratio, or relative risk) would be more interpretable.

Again, thank you for this observation. During our analysis, we had in fact run this logistic regression both using this standard cut-off of 2 and using alternative cut-offs (due to the high proportion considered distressed using the cut off of 2). We had decided not to use the logistic regression as combining the data in such a way reduces the data, cut-offs may be considered arbitrary and we found similar results using both the binary variable in the logistic regression and the continuous variable in the linear mixed regression. However, we have now included our logistic regression as a supplementary table and included reference to it in the results section.

In the discussion, the authors mention the self-selection bias in this study. Other than increasing the sample size, I think the authors could consider other bias-reduction techniques in this study.

 This has now been addressed in the limitations section.

  1. Support from government and requirement/ of amended responsibilities during the training, and guideline for practice during COVID could vary by country. These are also potential confounder.  However, it was neither discussed nor acknowledged as a limitation.

The study did not collect actual measures implemented in each country and hence what we measured was the respondent’s perception of support/responsibility change etc. We have edited each instance to ensure this is clear. Additionally, we have added a reference to this in the limitations. Additionally, references regarding the impact of such factors are discussed with relevant references in the discussion section.

Minor concerns:

  1. Please spell out the full name of PRICOV-19 when the first time it appeared. – We have done this in the first paragraph of the methods section where PRICOV-19 first mentioned in the paper.
  2. How many subjects with missing value of key covariates was not described at all. – The covariates are described in Tables 1 and 2. Table 2 already included the responding n for each factor; Table 1 had the n for each response category but we have added the n for each factor at the top now also for total clarity regarding the responding number for each variable.
  3. When talking about ICC, the authors mentioned “country clustering”, which confuses me. In my opinion, by “country clustering” we mean to sample several countries among all available countries – Apologies for any confusion cause in this regard – we were not referring to the sampling frame but to the fact that the responding practices are clustered in countries. We have edited the wording to be more accurate in this regard.
  4. The response variable eWBI score is integer-valued with a range -2 to 9. Further analysis may be needed to clarify why a linear mixed effect model, which is supposed to be applied for continuous variable, is appropriate for this integer-valued response variable.

Using a linear mixed model approach is a standard approach for this type of data with such sample size and that way we make use of all information. Additionally, we have re-checked and all relevant conditions for the approach were met and have now confirmed this in the methods - we checked for normality of residuals and for constant error variance (by plotting residuals against fitted values). In response to an earlier comment, we have also included the corresponding logistic regression in the supplementary material, which shows consistent findings.  

  1. The boxplots in Figure 1 shows skewness for some countries, for example, Austria and Denmark. This is again an evidence that the appropriateness of the linear mixed effect model needs to be validated.

We have re-checked and all relevant conditions for the approach were met and have now confirmed this in the methods. Additionally, in response to another comment, we have now also included reference to the country data in terms of variation and grouping. 

Reviewer 3 Report

The paper described the frequency of distress and well-being among general practitioners/family physicians during the COVID-19 pandemic and identified levers to mitigate the risk of distress. The paper was generally well written and could add value on existed literature and inform policy to make the health system more sustainable during the pandemic. The data and analysis generally supported the main conclusion.

I have a few comments and recommendations, which should be addressed further before it is accepted for publication.

  1. Page 3, 1st paragraph, line 4 Sampling and recruitment

…pre-defined recruitment procedure [10]. PRICOV-19 aimed to sample between 80 and 200 practices per country, depending on the total number of practices.

Please provide more details about the pre-defined recruitment procedure.

As illustrated in Supplementary STable 1, some countries (e.g. Israel [n=38], Sweden [n=18]) did not achieve the aimed sample size. Please discuss the limitations and the potential impact on the main results

  1. Page 3, Data analysis and further results

In Figure 1, it is clear that there is a huge country-level variation in eWBI score. Certain institutional factors of the health system (e.g. payment system etc) is known to affect GP’s well-being and mental health. In the method, the null model (Model 1) showed the country clustering of the data.

In addition to some countries have few respondents (or participants) as mentioned in point 1, it is worthwhile to present the country cluster from the model. The cluster may reflect health system characteristics, location etc. (e.g. North Europe, Eastern Europe etc.). Please also provide the justifications for the grouping.

  1. Page 9-10, Discussion –Limitations of the study

As acknowledged by the authors, the survey was “based on a self-selecting sample”. Please further discuss the potential bias of results due to this sampling strategy.

Please also provides a brief summary to describe “ the limitations and strengths of the PRICOV-19 study” in the discussion rather than simply citing a reference.

Author Response

 Page 3, 1stparagraph, line 4 Sampling and recruitment

…pre-defined recruitment procedure [10]. PRICOV-19 aimed to sample between 80 and 200 practices per country, depending on the total number of practices.

 Please provide more details about the pre-defined recruitment procedure.

We have added additional detail to the Sampling/Recruitment section and also expanded the supplementary table to include the recruitment/sampling strategy used in each country and the related response rate.

As illustrated in Supplementary STable 1, some countries (e.g. Israel [n=38], Sweden [n=18]) did not achieve the aimed sample size. Please discuss the limitations and the potential impact on the main results

We have included response rates in STable1 and included a commentary in the limitations section, which we hope you will agree addresses this point.

  1. Page 3, Data analysis and further results

In Figure 1, it is clear that there is a huge country-level variation in eWBI score. Certain institutional factors of the health system (e.g. payment system etc) is known to affect GP’s well-being and mental health.

We agree there is variation by country and Model 1 incorporates this and reports on the variation explained by country variation. We have edited the text in the methods and in the reporting of the results on this to ensure it is clear. We agree that there are factors such as the country payment structure etc that may impact on wellbeing. The questionnaire asked about the respondents’ perception of government support and this is reported on and included in the model. This is proxy for support only as the questionnaire did not collect data on this or on any new measures in place at the time of the survey. This has been referenced also under the limitations.

In the method, the null model (Model 1) showed the country clustering of the data. In addition to some countries have few respondents (or participants) as mentioned in point 1, it is worthwhile to present the country cluster from the model. The cluster may reflect health system characteristics, location etc. (e.g. North Europe, Eastern Europe etc.). Please also provide the justifications for the grouping.

We included individual country as a potential determinant, and we see while in subsequent models, the level of variance explained by country reduces, it remains significant. As outlined, above, details on country payment systems, actual supports etc was not collected, however, the practice and contextual variables do address many of these potential related aspects, albeit in a proxy format in some instances. The authorship group discussed at length a variety of potential country grouping variables but were not content that any of these were sufficient as in some instances data on all countries was not available from consistent public sources and in other instances were purely geographic based with countries within each not consistent in terms of these factors i.e. payment system etc. Additionally, some geographic groupings resulted in only two countries in the group and both health system structure factors and number of responses did not warrant this as a valid grouping. We have edited the reporting of the country variation in the results to provide more clarity and referred to the above in the methods section providing references also to indicate the groupings considered and why not considered valid to use for this study/paper.

  1. Page 9-10, Discussion –Limitations of the study

As acknowledged by the authors, the survey was “based on a self-selecting sample”. Please further discuss the potential bias of results due to this sampling strategy.

This has now been addressed in the limitations section.

Please also provides a brief summary to describe “ the limitations and strengths of the PRICOV-19 study” in the discussion rather than simply citing a reference.

The limitations and strengths have been elaborated further and this reference excluded as the detail is now provided in this paper.

Round 2

Reviewer 2 Report

The authors have addressed the comments. The paper has been improved after revision, though there is still room for further improvement.